# Supplementing *Citrus aurantium* Flavonoid Extract in High-Fat Finishing Diets Improves Animal Behavior and Rumen Health and Modifies Rumen and Duodenum Epithelium Gene Expression in Holstein Bulls

**DOI:** 10.3390/ani12151972

**Published:** 2022-08-03

**Authors:** Montserrat Paniagua, Javier Francisco Crespo, Anna Arís, Maria Devant

**Affiliations:** 1Ruminant Production, IRTA (Institut de Recerca i Tecnologia Agroalimentàries), Torre, Marimon, Caldes de Montbui, 08140 Barcelona, Spain; mpaniagua@quimidroga.com (M.P.); anna.aris@irta.cat (A.A.); 2Quimidroga S.A., 08006 Barcelona, Spain; 3HealthTech Bio Actives, S.L.U., 08029 Barcelona, Spain; jcrespo@htba.com

**Keywords:** bulls, flavonoids, performance, behavior, rumen inflammation, bitter taste receptors

## Abstract

**Simple Summary:**

Today, the commitment of beef production systems is to enhance their sustainability. Therefore, improving animal welfare and improving feed efficiency are both important challenges. Citrus flavonoids are polyphenols that have previously shown promising effects on reducing feed intake and modulating animal behavior in bulls fattened under commercial conditions. Moreover, increasing fat during the finishing phase in beef cattle diets has been used to increase dietary energy content and to reduce the risk of rumen acidosis. Therefore, the present study supports the hypothesis that a high dietary fat content does not interfere in the mode of action of these flavonoids and that the positive effects on feed efficiency and animal behaviors should be observed. Nevertheless, the mode of action of flavonoids is not known; it is speculated that digestive tract health (rumen) and changes in the expression of behavior- and inflammation-related genes (rumen and duodenum) may be involved, and dietary fat supplementation may also affect, antagonistic to flavonoids, these mechanisms. However, the present study data support that citrus flavonoid supplementation in high dietary fat diets may be a good strategy to face current beef production system challenges such as improving feed efficiency and animal welfare.

**Abstract:**

One hundred and forty-six bulls (178.2 ± 6.64 kg BW and 146.0 ± 0.60 d of age) were randomly allocated to one of eight pens and assigned to control (C) or citrus flavonoid (BF) treatments (*Citrus aurantium*, Bioflavex CA, HTBA, S.L.U., Barcelona, Spain, 0.4 kg per ton of Bioflavex CA). At the finishing phase, the dietary fat content of the concentrate was increased (58 to 84 g/kg DM). Concentrate intake was recorded daily, and BW and animal behavior by visual scan, fortnightly. After 168 d, bulls were slaughtered, carcass data were recorded, and rumen and duodenum epithelium samples were collected. Performance data were not affected by treatment, except for the growing phase where concentrate intake (*p* < 0.05) was lesser in the BF compared with the C bulls. Agonistic and sexual behaviors were more frequent (*p* < 0.01) in the C than in the BF bulls. In the rumen epithelium, in contrast to duodenum, gene expression of some bitter taste receptors (7, 16, 39) and other genes related to behavior and inflammation was higher (*p* < 0.05) in the BF compared with the C bulls. Supplementing citrus flavonoids in high-fat finishing diets to Holstein bulls reduces growing concentrate consumption and improves animal welfare.

## 1. Introduction

Citrus fruits contain a wide range of flavonoids. These flavonoids are polyphenols, a category of phytochemicals with plenty of biological activities, such as anti-inflammatory, antioxidant, and antimicrobial properties [1]. In previous research carried out with an extract from bitter orange (*Citrus aurantium*) rich in naringin (Bioflavex CA, HTBA, S.L.U., Barcelona, Spain), citrus flavonoids supplementation modified the eating pattern of Holstein bulls fed high-concentrate diets throughout the fattening period, reducing the large meal sizes consumed by the animals [2]. Furthermore, bulls supplemented with citrus flavonoids devoted more time to eat concentrate and straw and performed more ruminating activity during the finishing phase [2,3,4]. In the first study, a single-space feeder was used in order to study the eating pattern of the animals [2], and it was probably limiting the access to the concentrate, inasmuch as the use of a multiple-space feeder allowed bulls supplemented with citrus flavonoids to achieve the same performance as nonsupplemented animals [4]. Moreover, when the concentrate was fed in meal form, although the final BW and concentrate intake were not affected by treatment, the feed conversion ratio was improved in bulls supplemented with flavonoids [3]. Additionally, flavonoid supplementation reduced the gene expression of all bitter taste receptors (TAS2R) studied when concentrate was fed in meal form [3]. On the contrary, the gene expression of some of these genes was increased when the concentrate was in pellet form [4]. The expression of these different genes could be related to gut–brain axis mechanisms [3,4]. Consequently, as citrus flavonoid may interact with the digestive tract microbiota and the digestive tract receptors, their effect may be modified by feeding method (mash or pellet) modifying nutrient-sensing and other gut–brain crosstalk mechanisms in the rumen wall. Moreover, citrus flavonoids or their metabolites may also arrive to the small intestine and modify the nutrient-sensing mechanisms in that site. The use of feed additives to modulate the eating and animal behavior is an increasing area of interest and research. Traditionally, increasing dietary fat content at the finishing phase has been a strategy to fulfill animals energy requirements to modulate feed intake and reduce the meal size [5,6,7] and to control bloat and rumen acidosis problems. Furthermore, an increase in blood concentration of cholecystokinin (CCK) and pancreatic polypeptide (PP) has been related to the reduction in feed intake associated with high fat levels in the diet of cows [8]. Actually, PP is considered a member of the NPY family, composed by neuropeptide Y (NPY), peptide YY (PYY) and PP, acting all of them upon the same family of receptors, NPYR [9,10]. Bulls supplemented with flavonoids in a concentrate in meal form have showed a decline in the gene expression of receptors related with these neurotransmitters and hormones, such as CCKBR (acting as CCK and gastrin receptor) [11] and PPYR [3]. Thus, the effects observed in cattle when high-fat diets are used might be similar to those observed when the concentrate has been supplemented with citrus flavonoids, reducing meal size and concentrate intake. Additionally, in both cases, metabolic pathways involving CCK and NPY family are likely playing a key role. Consequently, under commercial conditions in beef cattle, the possible existence of an interaction (antagonistic effects) between dietary fat and citrus flavonoid supplementation in high-concentrate diets should be studied.

Finally, oral non-nutritive behaviors, agonistic interactions (fighting, butting, and chasing) and sexual behaviors (flehmen, attempted and complete mounts) were also reduced in bulls supplemented with flavonoids independently of the feeder or concentrate presentation (pellet or meal) [2,3,4]. Our previous studies showed that flavonoid supplementation modulated the expression of some genes related to the gut–brain axis in the rumen of bulls, although these results were affected by the presentation form of the concentrate, pellet [4] or meal [3]. As mentioned previously, citrus flavonoids or their metabolites may also arrive to the small intestine and modify the nutrient-sensing mechanisms in that site. These crosstalk mechanisms taking place in the digestive tract, including the rumen or the small intestine, as proposed in monogastric animals, have been poorly studied in ruminants.

Thus, the present study was designed to evaluate the effects of citrus flavonoid supplementation on concentrate consumption, growth rate, concentrate efficiency, macroscopic rumen wall health, carcass characteristics, and animal behavior in Holstein bulls fed high-fat concentrate diets in commercial conditions. Furthermore, the present study also aimed to study deeper how citrus flavonoid supplementation in high-fat concentrate diets could affect the expression of some genes involved in gut–brain crosstalk mechanisms in the rumen and duodenum epithelium, such as bitter taste receptors and inflammation regulators.

## 2. Materials and Methods

### 2.1. Animals, Feeding, Housing, and Experimental Design

This study was conducted in accordance with the Spanish guidelines for experimental animal protection (Royal Decree 53/2013 of February 1st on the protection of animals used for experimentation or other scientific purposes; Boletín Oficial del Estado, 2013). One hundred and forty-six Holstein bulls (178.2 ± 6.64 kg BW and 146.0 ± 0.60 d of age) were fattened under commercial conditions in a farm (Granja l’Alsina, L’Alsina, Lleida, Spain). The study was divided into growing (0 to 112 d) and finishing (113 to 168 d) phases, after 168 animals were fed the same treatments until slaughter day. During the study, animals were not vaccinated. Animals were randomly allocated in one of eight pens and assigned to one of the two treatments (4 pens per treatment and 18 animals/pen), either control (C) or supplemented (BF) with 0.04% of bitter orange extract (*Citrus aurantium*) of the whole fruit, rich in naringin (24%) (Bioflavex CA, HTBA, S.L.U., Barcelona, Spain). Bioflavex was incorporated into the concentrate during the concentrate manufacturing. Concentrates were manufactured from a 9000 kg master batch, of which 4500 kg were C, and the other 4500 kg BF. Each treatment concentrate was transported to the farm with the same truck and stored into two different silos under the same conditions.

Pens were totally covered (12 m × 6 m), deep-bedded with straw, and equipped with a three-space feeder (1.50 m length, 0.40 m width, 1.50 m height, and 0.35 m depth). The feeder of each pen weighed the concentrate continuously as described by Verdú et al. [12], and these data were recorded to calculate concentrate consumption by pen. Pens were also equipped with one drinker (0.30 m length, 0.30 m width, 0.18 m depth). Straw was offered ad libitum in a separated straw five-space feeder (3.60 m length, 1.10 m wide, and 0.32 m depth), and every time it was replaced, it was recorded to estimate the total straw consumption. As straw was also used for bedding, these data were only an estimation.

### 2.2. Feed Consumption and Performance

Animals were fed a commercial concentrate in meal form, formulated to cover their nutritional requirements [13]. The first 112 d of the study, animals were fed a grower concentrate formula, and between 112 d to the end of the study, animals were fed a finisher concentrate with a high fat content (around 84 g/kg DM), which is used in some commercial feedlots that feed mash at the finishing stage to increase the energy content and achieve an increased carcass fatness score and reduced risk of rumen acidosis. Ingredients and nutritional composition of the concentrates are shown in Table 1. Throughout the study, animals had ad libitum access to wheat straw (3.5% CP, 1.6% ether extract, 70.9% NDF, and 6.1% ash; DM basis) and fresh water. Animals were weighed individually every 14 d throughout the study in 12 experimental periods of 14 d.

### 2.3. Animal Behavior

A visual scan procedure at days 13, 28, 44, 56, 72, 83, 100, 114, 128, 143, 153, and 167 of the study was performed to study the general activity (standing, lying, eating, drinking, and ruminating) and social behavior (nonagonistic, agonistic, and sexual interactions) of the animals in every pen. The visual observation was made for 2 pens at the same time from 8:00 to 10:30 a.m., as described by Verdú et al. [14]. General activities were scored using 3 scan samplings of 10 s at 5 min intervals, and social behavior was scored during 3 continuous sampling periods of 5 min. This scanning procedure of 15 min was repeated twice consecutively in each pen, starting randomly in a different pen every scanning day. This method describes a behavior exhibited by an animal at a fixed time interval [15].

### 2.4. Carcass Quality

After 168 d of study, the bulls were transported to the slaughterhouse (Escorxador del Grup Alimentari Guissona, Guissona, Spain), located 15 km from the farm. Animals were slaughtered in 2 weeks, four pens per week, two pens from the control and two from the BF bulls each week. The time spent waiting before slaughter was less than 6 h. Animals were weighed before loading. They were slaughtered by commercial practices and following the EU Regulation 1099/2009 on the protection of animals at the time of killing or slaughtering. After slaughtering, the hot carcass weight (HCW) was registered for every animal. The dressing percentage was calculated by dividing the HCW by the BW recorded before slaughtering. Moreover, following the (S) EUROP categories described by the EU Regulation nos. 1208/81 and 1026/91, the conformation of carcasses was classified, where “E” corresponded to an excellent conformation, “U” to very good conformation, “R” to good conformation, “O” to fair conformation, and “*p*” to a poor conformation. The fat cover was classified according to the EU Regulation no. 1208/81, which utilizes a classification system by numbers, 1, 2, 3, 4, 5, where 5 explains a very high degree of covering fat and heavy fat deposits in the thoracic cavity, and 1 is classified as a low degree, with no fat cover.

### 2.5. Rumen and Liver Macroscopic Evaluation and Sample Collection

All animals’ rumen and the liver of every animal were macroscopically evaluated at the slaughterhouse. Rumens were classified depending on the color by a visual evaluation, from 1 to 5, “5” being a black-colored rumen and “1” a white-colored rumen [16]. They were also divided into areas according to Lesmeister et al. [17] to examine the presence of ulcers, baldness regions, and clumped papillae [18]. Liver abscesses were classified according to Brown et al. [19].

Additionally, a liquid sample from the rumen was obtained from homogeneous contents strained with a cheesecloth from 18 animals per treatment randomly selected from two pens per treatment, immediately following slaughter. Following the procedures of Jounay [20], 4 mL of ruminal fluid was mixed with 1 mL of a solution containing 0.2% (wt/wt) mercuric chloride, 2% (wt/wt) orthophosphoric acid, and 2 mg/mL of 4methylvaleric acid (internal standard) in distilled water, and stored at −20 °C until subsequent VFA analysis. Furthermore, a 1 cm^2^ section of rumen wall (left side of the cranial ventral sac) and duodenum epithelium was sampled. Ruminal papillae from the rumen wall section was excised, and both ruminal papillae and duodenum epithelium samples were rinsed 2 times with chilled PBS after sampling and immediately incubated in RNAlater (Invitrogen, Madrid, Spain) to preserve the RNA integrity. After 24 h of incubation with RNA later at 4 °C, the liquid was removed, and the tissue was frozen at −80 °C until further RNA extraction and subsequent gene expression analysis.

### 2.6. Biological and Chemical Analyses

During the study, samples of concentrate were collected at days 0, 42, 84, 126, and 168 and analyzed for DM [21], ash (method 642.05) [21], CP by the Kjeldahl method (method 988.05) [21], ADF and NDF according to Van Soest et al. [22] using sodium sulfite and alpha-amylase, and EE by Soxhlet with a previous acid hydrolysis (method 920.39; AOAC, 2005) [21].

Naringin was determined for every sample of concentrate (C and BF) as a Bioflavex CA marker for the BF group and was used as a marker confirming the adequate inclusion of citrus flavonoid extract in the diets by Laboratory of Interquim S.A. An internal method for naringin quantification using HLPC developed by Interquim S.A. was used [2].

Ruminal VFA concentration was determined with a semicapillary column (15 m × 0.53 mm ID, 0.5 µm film thickness, TRB-FFAP, Teknokroma, Barcelona, Spain) composed of 100% polyethylene glycol (PEG) esterified with nitroterephtalic acid, bonded and crosslinked phase (method number 5560; APHA–AWWA– WPCF, 2005), using a CP-3800-GC (Varian, Inc., Walnut Creek, CA, USA).

For the gene expression analyses, total RNA was extracted from rumen papillae and duodenum epitheliums homogenizing tissues in Trizol (Invitrogen, Carlsbad, CA, USA) by Polytron Instrument (IKA, Staufen, Germany). Isolated mRNA was reverse-transcribed to cDNA using a PrimeScript RT Reagent Kit (Takara, Frankfurt, Germany) following the manufacturer’s instructions. The RNA purity was assessed by a NanoDrop instrument (ThermoFisher, Madrid, Spain) at 260, 280, and 230 nm. The criteria used to consider a good RNA quality was a ratio 260/280 and 260/230 in the range of 2.0 and 2.0–2.2, respectively. The quantification of the expression of genes at the mRNA level coding for: (1) the tight-junction protein claudin 4 (CLDN4); (2) the production, expression, and turnover of neurotransmitters: free fatty acid receptor 2 (GPR43) and free fatty acid receptor 3 (GPR41), pancreatic polypeptide receptor 1 (PPYR1); actual name neuropeptide Y receptor Y4 (NPY4R), α2-adrenergic receptor subtype C (ADRA2C), and cholecystokinin receptor 4 (CCKBR); (3) proinflammatory cytokines TNF-a (TNFa) and cytokine IL-25 (IL-25), pattern recognition receptor Toll-like receptor 4 (TLR4), and antimicrobial peptides released by intestinal cells (beta-defensins and lactoferrin); (4) bitter taste receptors type 2 members 7, 16, 38, and 39 (TAS2R7, TAS2R16, TAS2R38, and TAS2R39) were performed by quantitative PCR (qPCR). The qPCR was performed using gene codifying for ribosomal protein subunit 9 (RPS9) as a housekeeping gene, which was checked for stability following Vandesompele et al. [23] in comparison with genes codifying for b-actin (ACTB), ubiquitously expressed transcript protein (UXT) and glyceraldehyde 3-phosphate dehydrogenase (GAPDH). The qPCR conditions for each set of primers were individually optimized [3]. The specificity of the amplification was evaluated by single-band identification at the expected molecular weight in 0.8% DNA agarose gels and a single peak in the melting curve. The efficiency was calculated by amplifying serial 1:10 dilutions of each gene amplicon. A standard curve of crossing point (Cp) versus the logarithm of the concentration was plotted to obtain the efficiency, which was calculated using the formula 101/slope, with an acceptable range of 1.8 to 2.2. A total reaction volume of 20 μL was used, containing 50 ng of cDNA, 10 μL of SYBR Premix EX Taq (TliRNAseH) (Takara, Frankfurt, Germany) and the optimized primer concentration for each gene [3]. The qPCR reactions were performed as follows: an initial denaturing step of 10 min at 95 °C followed by 40 cycles of 10 s at 95 °C, 15 s at the optimized annealing temperature for each gene, 30 s at 72 °C, and a final extension of 10 min at 72 °C. The resulting Cp values were used to calculate the relative expression of selected genes by a relative quantification using a reference gene (housekeeping gene) and a calibrator of the control group ([24], Equation (3.5)).

### 2.7. Calculations and Statistical Analyses

Only the pen was considered the experimental unit and animals within the pen were considered sampling units in some parameters.

The concentrate efficiency data were log-transformed to achieve a normal distribution. The means presented in the tables and figures correspond to nontransformed data and the SEM and *p*-values correspond to the ANOVA of the transformed data. daily. Then, these data were transformed into natural logarithms to achieve a normal distribution. The frequency of each social behavior was calculated by summing by day, pen, and scan, and transformed into the root of the sum of each activity plus 1 to achieve a normal distribution. The ANOVA was performed with transformed data, and the means shown in the tables correspond to the backtransformed data.

The unification of performance, animal behavior, and concentrate consumption data averaged by pen and period were analyzed using a mixed-effects model (Version 9.2, SAS Inst., Inc., Cary, NC, USA). The model included the initial BW as a covariate, the treatment, period (14 d period), and the interaction between treatment and period as fixed effects, and the interaction between treatment and pen and the 3-way interaction between treatment, pen, and period as random effects. The period was considered a repeated factor, and for each analyzed variable, the animal nested within the interaction between treatment and pen (the error term) was subjected to 3 variance–covariance structures: compound symmetry, autoregressive order one, and unstructured. The covariance structure that yielded the smallest Schwarz’s Bayesian information criterion was considered the most desirable analysis.

In the case of the rumen gene expression and VFA data, the pen was considered the experimental unit and the animals the sampling units, and data were analyzed using ANOVA where the model included treatment (as there were no repeated measures) as the main effect. For the categorical variables analyses (carcass classification, rumen health parameters, hepatic abscesses) a Chi-squared test was used.

Differences were declared significant at *p* < 0.05, and trends were discussed at 0.05 ≤ *p* ≤ 0.10 for all models.

## 3. Results

### 3.1. Animal Health

Two animals did not finish the study, one from the C treatment due to enterotoxaemia and one from the BF treatment due to chronic respiratory problems. 

### 3.2. Intake, Performance, and Carcass Quality

During the growing phase, concentrate intake was greater (*p* < 0.05) for the C than for the BF bulls, whilst these differences disappeared for the finishing phase (Table 2). Furthermore, no statistical differences between treatments were found for this parameter when analyzing the whole study (Table 2). Moreover, the estimation for straw consumption did not show statistical differences between treatments during the growing phase, being 0.54 ± 0.021 kg/d for the C bulls and 0.52 ± 0.021 kg/d for the BF animals, and neither for the finishing phase, when straw intake was 1.02 ± 0.132 kg/d and 0.76 ± 0.132 kg/d for the C and BF bulls, respectively.

On the other hand, the ADG, final BW, and FCR were not affected by the treatments either during the growing phase (Table 2) nor during the finishing phase or when analyzing the whole study together (Table 2). At the slaughterhouse, the BW, dressing percentage, carcass conformation, and fatness classification were not affected by the treatments (Table 3).

### 3.3. Animal Behavior

All data for animal behavior, including both general activities and active behavior, are shown in Table 4 for the growing phase and Table 5 for the finishing phase.

#### 3.3.1. General Activities

In the growing phase, no statistical differences were found in the percentage of animals per pen standing, lying, and drinking water during the visual observation period. Otherwise, the percentage of animals eating concentrate and eating straw were greater (*p* < 0.01) for the BF compared with C bulls, and the proportion of animals ruminating tended (*p* < 0.10) to be greater as well for the BF bulls than for the C bulls in this phase. During the finishing phase, as in the growing phase, no differences during the visual observation period were found in the proportion of animals per pen standing, lying, and drinking water, and in this phase, between treatments, no differences in animals eating straw. As observed in the growing phase, in this phase, the proportion of animals eating concentrate and ruminating were greater (*p* < 0.01) in the BF bulls compared with the C bulls.

#### 3.3.2. Active Behavior

In the growing phase, during the visual scan observation period, self-grooming and social behavior were not affected by the treatment. On the other hand, oral non-nutritive behaviors were more frequently (*p* < 0.01) performed by the C compared with the BF bulls, as well as all agonistic behaviors (fighting, butting, chasing, and chasing-up) except displacement were also more frequently (*p* < 0.05) exhibited by the C bulls than the BF bulls. Regarding sexual behaviors, flehmens and attempts to mount were more frequent (*p* < 0.05) in the C compared with the BF bulls, although complete mounts were not affected by the treatments. During the finishing phase, no differences between treatments were observed for social behavior, whilst the bulls from the BF group performed (*p* < 0.05) more self-grooming than the C bulls. Additionally, during this phase the C bulls also performed more (*p* < 0.01) oral non-nutritive behaviors than the BF bulls. Again, differences among treatments in agonistic behaviors were found in this phase. Fighting, butting, and displacement behaviors were more frequent (*p* < 0.01) in the C bulls compared with the BF bulls, and the C bulls also tended (*p* < 0.10) to show more chasing-up behaviors than the BF animals. In regard to sexual behaviors, the C bulls performed more (*p* < 0.01) flehmen behaviors and tended (*p* < 0.10) to perform more attempted and complete mounts compared with the BF bulls.

### 3.4. Macroscopic Rumen Evaluation and Liver Abscesses

The results of the macroscopic evaluation at the slaughterhouse of rumens and livers are shown in Table 6. The color of the rumen wall was lighter (*p* < 0.01) in the BF bulls (72.22% classified as color <“3”) compared with the C bulls (46.58% classified as color <“3”). The number of baldness regions in the rumen also tended (*p* < 0.10) to be greater in the C bulls (16.44%) compared with the BF bulls (6.94%). In the remaining macroscopic parameters analyzed at the slaughterhouse (liver abscesses, ulcers, and clumped papillae) no differences among treatments were found.

The data of ruminal liquid parameters analyzed are presented in Table 7. The rumen pH was greater (*p* < 0.01) in the BF than in the C bulls, and on the contrary, the total VFA concentration was greater (*p* < 0.01) for the C bulls compared with the BF bulls. The molar proportion of acetate was also affected by the treatments, being greater (*p* < 0.01) in the BF bulls compared with the C bulls, whereas the molar proportion of propionate was greater (*p* < 0.01) for the C bulls than for the BF bulls. Furthermore, the butyrate molar proportion was also greater (*p* < 0.05) in the BF than in the C bulls. The remaining VFAs analyzed (valerate, isobutyrate, and isovalerate) were not affected by the treatment. As for the acetate and propionate molar proportions, the acetate:propionate ratio was greater (*p* < 0.05) for the BF bulls than for the C bulls. 

### 3.5. Expression of Genes in the Rumen and Duodenum Epithelium

The data of the relative gene expression in the rumen epithelium are presented in Figure 1. The supplementation with flavonoids affected the expression of all bitter taste receptors (TAS2R) analyzed except TAS2R38. The relative expression of TAS2R7, TAS2R16, and TAS2R39 was greater (*p* < 0.01) in the rumen of the BF compared with the C bulls. The relative expression of all receptors related with the neurotransmitter signaling studied (FFAR3and FFAR2, ADRAC2C, PPYR1, and CCKBR) was greater (*p* < 0.05) in the BF bulls compared with the C bulls. Furthermore, the relative expression of some receptors related with inflammation such as IL-25, TLR4, and defensin, was also greater (*p* < 0.05) for the BF bulls than for the C bulls.

Data of the relative expression of duodenum epithelium genes are presented in Figure 2. Again, the supplementation with flavonoids affected the expression of all bitter taste receptors (TAS2R) analyzed except TAS2R38. Contrary to the results in the rumen epithelium, in the duodenum, the relative expression of TAS2R7, TAS2R16, and TAS2R39 was greater (*p* < 0.001) in the C bulls compared with the BF bulls. The relative expression of some receptors related with the neurotransmitter signaling also differed among treatments. The FFAR2 expression (*p* < 0.01) was more expressed in the C bulls compared with the BF bulls. Furthermore, the relative expression for PPYR1 and CCKBR was greater (*p* < 0.01) as well for the C bulls than for the BF bulls. Additionally, the relative expression of the receptors related with inflammation, such as IL-25, TLR4, and defensins, were again greater (*p* < 0.05) for the C bulls than for the BF bulls, contrary to the rumen epithelium results.

## 4. Discussion

Citrus flavonoid supplementation clearly reduced concentrate intake in bulls during the growing phase, although the remaining performance parameters analyzed were not affected throughout the study. Moreover, when the bulls were supplemented with citrus flavonoids, they spent more time eating concentrate and straw than the C bulls, and also devoted more time to performing ruminating activities as observed during the visual scan procedure for the growing phase. In all our previous studies [2,3,4], bulls supplemented with citrus flavonoids devoted more time to eating concentrate during the growing phase and with the exception of one study [4], a decrease in concentrate intake was also observed. Potential mechanisms whereby this eating regulation occur have been previously discussed [2,3,4]: (i) increasing the propionic acid production and reducing the acetate:propionate ratio in the ruminal liquid, affecting the eating pattern of bulls [2,3]; (ii) modifying the gene expression of bitter taste receptors (TAS2R) and anorexigenic peptides and hormones in the rumen epithelium [3,4]; (iii) modulating animal behavior through mechanisms involved in the gut–brain axis, such as inflammation [2,3,4].

In the present study, the fat content was increased during the finishing phase; this strategy is commonly used to increase energy density in finishing diets of fattening bulls [25,26]. However, high-fat diets modify eating patterns in cattle, and supplementing fat above 6% to 7% may reduce feed intake [25,27]. Furthermore, the type of fat and the degree of saturation of this fat can affect the extent of reduction in the feed intake of supplemented diets [28]. Additionally, satiety hormones such as cholecystokinin (CCK) and pancreatic polypeptide (PP) have been associated with a reduction in feed intake related to high fat levels in cows [8,28]. Previously, citrus flavonoid supplementation was shown to decrease the gene expression of cck-br and ppyr1 in the rumen epithelium [3]. The release of these anorexigenic hormones, such as CCK, neuropeptide Y (NPY), and peptide YY (PYY), are triggered by the activation of bitter taste receptors (TAS2R) [29,30,31]. Thus, it was expected that as citrus flavonoid would decrease the gene expression of bitter taste receptors, this in turn could have caused the decrease in the gene expression of CCKBR and PPYR1 in the rumen epithelium. When feeding fat, the effect of citrus flavonoid supplementation could counterbalance the effect of fat supplementation on the gene expression of CCKBR and PPYR1 in the rumen epithelium, so a possible interaction may occur when supplementing citrus flavonoids in high-fat concentrate in bulls as mentioned in the introduction. However, quite the opposite of our previous study [3], citrus flavonoids, when fed with high-fat (palm oil) diets, sharply increased the relative gene expression of almost all TAS2R analyzed in the rumen epithelium of bulls fed a high-fat diet during the finishing phase. These results raise different questions; the first question is if a fat diet (palm oil) is fed, why is citrus flavonoid supplementation exerting the opposite effect in the expression of the genes analyzed in the rumen epithelium? One explanation could be that fats, and especially oils (in this study palm oil was used), exert negative effects on ruminal microflora growth, particularly affecting protozoa and fibrolytic bacteria [32]. Naringin, a bitter-tasting glycosylated flavanone, is rapidly deglycosylated to naringenin by the rumen microflora [33,34], and naringenin acts as an important bitter-masking molecule [35]. Cheng et al. [36] found that strains of ruminal *Butyrivibrio* spp. hydrolyzed the glycosidic bond, metabolizing naringin to naringenin in rumen. Interestingly, recent research has shown that *Butyrivibrio* spp. growth was clearly inhibited by the presence of oils and fats at low concentrations [32]. Consequently, it might be hypothesized that high-fat diets might interfere with the deglycosylation of naringin to naringenin by rumen bacteria such as *Butyrivibrio* spp., that would explain the higher gene expression of TAS2R in ruminal epithelium of bulls supplemented with citrus flavonoids. Due to this increase in the gene expression of TAS2R, the production, expression, and turnover of neurotransmitters studied were probably also more expressed in the bulls supplemented with citrus flavonoids.

The differences in the expression of the following genes would support a decreased feed intake: (i) as previously mentioned, the activation of TAS2R triggers the release of anorexigenic molecules, such as CCK and PYY [31,37,38], and fatty acids are also related to an increase in CCK and PP in the gastrointestinal tissues of ruminants [8,28]. (ii) Moreover, in this study, the gene expression of ADRAC2C in the rumen epithelium of the BF bulls was greatly increased, and this receptor induced a reduction in forestomach contractions [39], which could lead to a reduction in concentrate intake, reducing the passage of ruminal content to the abomasum and intestines. (iii) On the other hand, specific nutrient-sensing receptors for fats, the free fatty acid receptors (FFAR), have been described throughout the digestive tract of different animal species [40]: FFAR1 (GPR40), activated by long- and medium-chain fatty acids; FFAR2 (GPR43), activated by short-chain fatty acids; FFAR3 (GPR41), also activated by short-chain fatty acids; and, finally, FFAR4 (GPR120), activated specifically by long-chain fatty acids. All these FFAR are found in enteroendocrine cells and are responsible for the secretion of different anorexigenic molecules such as glucagon-like peptide-1 (GLP-1), PYY, and CCK [37,40]. Our results showed an increase in FFAR2 and FFAR3 gene expression in the bulls supplemented with citrus flavonoids, whilst VFA analyzed at the slaughterhouse clearly exhibited a reduction in the total VFA concentration in the BF bulls and a reduction in molar proportions of the propionate as well. Recent research has demonstrated interactions between bitter and lipid sensing [38], so it could be hypothesized that citrus flavonoids supplementation to bulls fed high-fat diets might increase the gene expression of FFAR2 and FFAR3 due to an interaction among bitter and fat nutrient sensing pathways. Actually, bitter taste and fat seem to be both determinant pathways playing a key role in satiety mechanisms in the animals. In fact, bitter taste could be considered as one of the main sensory defense mechanisms in animals to avoid the ingestion of toxic substances, so it makes sense that the activation of TAS2R throughout the digestive tract triggers the activation of anorexigenic molecules and pathways sending signals to the brain to stop eating. On the other hand, fats are the nutrients with the highest energy value, triggering also important cues activating satiety in the brain. Indeed, both the supplementation of bitter taste additives and fats have been shown to reduce the meal size in bulls [2,25,27]. Consequently, supplementing citrus flavonoids to bulls fed high-fat diets could exacerbate the release of anorexigenic hormones and peptides, explaining the results observed in the gene expression in the rumen epithelium. An additional question would be: if gene expression in the rumen epithelium of genes related with eating pattern was so different among treatments (Figure 1) during the finishing phase, why were no differences among treatments in feed intake (amount) observed during this phase? May be the answer is related to the gene expression in the duodenum epithelium, where in contrast to the rumen epithelium, for almost all nutrient-sensing receptors studied, gene expression was reduced in the bulls supplemented with citrus flavonoids compared with the C bulls, including TAS2R and receptors of anorexigenic peptides (such as FFAR2, CCKBR, ADRA2C, and PPYR1). Surprisingly, these results would be in agreement with the results observed in the rumen epithelium of our previous study [3], when the concentrate was not supplemented with this large quantity of fat (palm oil). Therefore, these results could support the hypothesis that in the present study naringin could have been deglycosylated to naringenin just before arriving or upon arriving at the duodenum, acting in this part of the intestine as a bitter-masking molecule and blocking the TAS2R and the anorexigenic molecules released by these receptors. Maybe the release of anorexigenic molecules in the rumen reduced forestomach contractions (CCK, PPY, and ADRA2C), increasing the time of passage and allowing naringin deglycosylation by rumen bacteria. Certainly, these changes in gene expression observed in the duodenum epithelium are also affecting feeding behavior of the bulls, along with the differences observed in the rumen epithelium. All these mechanisms have been described as important part of the gut–brain axis network regulating eating patterns in animals, but much more research is needed to properly describe the importance of the digestive tract site where those mechanisms may take place, if the effects are additive across the digestive tract sites, and how these changes in gene expression modulate hunger and satiety in bulls.

As previously mentioned, citrus flavonoid supplementation clearly decreased the total VFA in ruminal liquid at the slaughterhouse and increased ruminal pH. Moreover, the acetate molar proportion was increased and the propionate molar proportion reduced, so that the acetate:propionate ratio was higher for the BF bulls. Conversely, the gene expression of FFAR2 and FFAR3 was higher for the BF bulls as previously explained, probably due to bitter and lipid senses interaction, being then completely independent of the ruminal VFA content. Previous research carried out with citrus flavonoids also showed an improvement in ruminal pH probably as consequence of rumen microflora modulation, but in previous studies [41] the concentration of ruminal propionate was greater when citrus flavonoids were supplemented compared with the present one. These differences could be because the extract of citrus flavonoids and dose supplemented were different and, furthermore, sampling the ruminal liquid for the analysis at the farm or at the slaughterhouse can affect the ruminal VFA concentration and profile [42]. In the present study, this greater ruminal pH along with a low rumen VFA concentration could explain that the macroscopic rumen wall parameters studied at the slaughterhouse were lesser in the BF bulls. The rumen wall color was lighter and the number of baldness areas were reduced in the bulls supplemented with citrus flavonoids. These parameters might be indicative of better ruminal health and could also affect the VFA absorption through the rumen epithelium.

In the present study, in agreement with our previous research [2,3,4], citrus flavonoids supplementation modulated animal behavior, reducing oral non-nutritive behaviors and aggressive and sexual interactions. In beef cattle, oral non-nutritive behavior and aggressive interactions are indicators of poor welfare and stress [43,44], whereas sexual interactions, especially during the finishing phase, can negatively affect carcass quality and the well-being of animals due to different injuries. The microbiome gut–brain axis is an extensive communication network between the brain, the gut, and their microbiota [45]. Beyond its importance in nutrient homeostasis, the gut–brain axis has been suggested to be involved in behavior modulation and mood disorders [45,46]. Consequently, the diet, but also inflammation, has been proposed as one of the key factors modulating animal behavior through this gut–brain axis network [47]. In beef cattle, recent studies [3,4,6] have suggested that some of these crosstalk mechanisms modulating animal behavior could happen in the rumen. In agreement with our previous results [2,3,4], in the present study, the bulls supplemented with citrus flavonoids reduced their oral non-nutritive behaviors throughout the study. This stereotypic behavior has been related in beef cattle with digestive dysfunctions, such as low pH or ruminal lesions, or even with the impossibility of performing natural eating behaviors such as rumination [44]. As mentioned previously, our results have shown an improvement in macroscopical ruminal wall parameters such as a lighter color and a smaller presence of baldness areas, along with an increase in ruminal pH, when the bulls were supplemented with citrus flavonoids. Additionally, the BF bulls also performed more ruminating activities during the visual scan procedure. Therefore, in this study, citrus flavonoids might have reduced oral non-nutritive behaviors by improving ruminal health and increasing pH and natural behaviors such as rumination. In the present study again citrus flavonoids supplementation reduced all agonistic and sexual interactions, especially during the finishing phase. Gut–brain axis crosstalk, such as gastrointestinal inflammation, neuropeptides, but also nutrient sensing mechanisms, such as TAS2R and FFAR, have been proposed for this animal behavior modulation by citrus flavonoids supplementation [3,4]. In our previous results carried out with a concentrate in meal form [3], citrus flavonoids supplementation clearly reduced the gene expression of some proinflammatory molecules in the rumen epithelium (IL-25, TLR4, and defensin) and in some they increased [4]. Inflammation has been suggested to be involved in animal behavior modulation by decreasing serotonin in serum, an important neurotransmitter in the gut–brain axis associated with aggressive behaviors [45,47,48]. Surprisingly, in the present study, the gene expression of these proinflammatory molecules (IL-25, TLR4, and defensin) in the rumen epithelium showed exactly the opposite results, being more expressed in the BF bulls. Fatty acid excess has been related with inflammation through the activation of TLR-4, which is also activated by LPS and related to innate immune response, activating the secretion of proinflammatory cytokines [49,50]. Additionally, the gene expression in the rumen of the TAS2R analyzed was also higher in the BF bulls, and these receptors also play important roles in the immune response [51]. Consequently, this increase in proinflammatory molecules in the rumen epithelium when citrus flavonoids were supplemented in the bulls fed high-fat diets would be the result of the TAS2R interaction with FFAR, also triggering an exacerbated response. Contrary to the rumen observations, the gene expression of these proinflammatory molecules in the duodenum epithelium of the BF bulls was reduced, as observed in the rumen epithelium of our previous study [3]. These results were also in agreement with the reduction of the nutrient-sensing receptors found in the duodenum epithelium of the BF bulls, such as TAS2R, FFAR2, ADRA2C, PPYR1, and CCKBR. Thus, as previously discussed, maybe naringin is deglycosylated to naringenin just before or upon arriving at the duodenum, exerting its effects as a bitter-marking molecule and as a potent antioxidant. As in our previous studies [2,3,4], eating pattern modulation observed in bulls when supplemented with citrus flavonoids could reduce aggressive and abnormal behaviors by increasing time devoted to feeding events, such as eating concentrate, straw, and rumination.

In summary, animal and eating behavior results were similar to those of previous studies [2,3,4]; however, when bulls were fed citrus flavonoid and high-fat (palm oil) diets, the gene expression pattern was the opposite in the rumen epithelium than that observed in mash diets without high fat levels [3], but similar to pellet diets [4] and to the duodenum epithelium.

## 5. Conclusions

In conclusion, during the growing phase, feeding bulls with high-concentrate diets supplemented with citrus flavonoid decreased concentrate intake. Moreover, when bulls fed high-concentrate diets were supplemented with citrus flavonoids, bulls reduced oral non-nutritive behaviors, agonistic interactions, and sexual behaviors, potentially improving animal welfare. Ruminal pH and rumen wall parameters macroscopically studied suggested that rumen health was better in the BF than in the C bulls. Finally, flavonoid supplementation in a high-fat finishing diet differently modified the gene expression of genes in the rumen and duodenum epithelium, and further research is needed to relate these data with potential regulation mechanisms of eating pattern and animal behavior.

## Figures and Tables

**Figure 1 animals-12-01972-f001:**
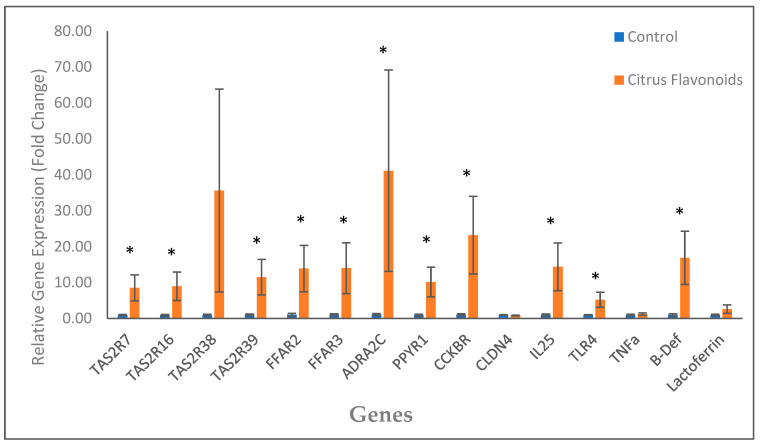
Gene expression in rumen epithelium of Holstein bulls fed high-concentrate diets with or without citrus flavonoids supplementation (* = *p* < 0.05). TAS2R7: *bitter taste receptor 7*; TAS2R16: *bitter taste receptor 16*; TAS2R38: *bitter taste receptor 38*; TAS2R39: *bitter taste receptor 39*; FFAR2: *free fatty acid receptor 3 (gpr41)*; FFAR3: *free fatty acid receptor 2 (gpr43)*; ADRA2C: *alpha 2 adrenergic receptors subtype C*; PPYR1: *pancreatic polypeptide receptor 1*; CCKBR: *cholecystokinin receptor 4*; IL-25: *interleukin-25*; TLR4: *pattern recognition receptors, such as Toll-like receptor 4*; TNFa: *tumor necrosis factor alpha*; B-Def: *beta-defensin*.

**Figure 2 animals-12-01972-f002:**
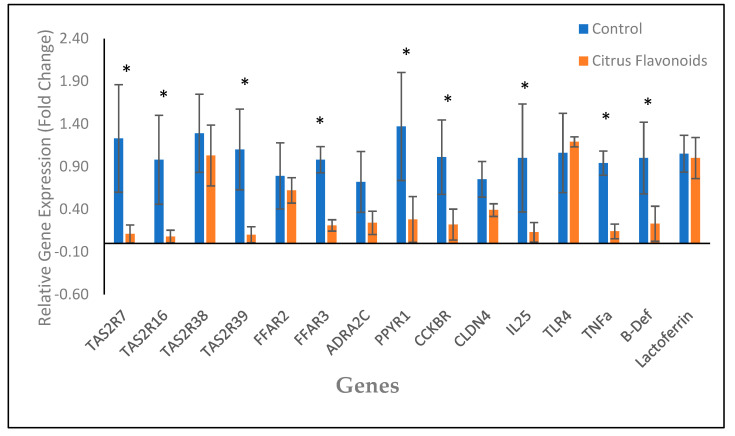
Gene expression in duodenum of Holstein bulls fed high-concentrate diets with or without citrus flavonoids supplementation (* = *p* < 0.05). TAS2R7: *bitter taste receptor 7*; TAS2R16: *bitter taste receptor 16*; TAS2R38: *bitter taste receptor 38*; TAS2R39: *bitter taste receptor 39*; FFAR2: *free fatty acid receptor 3 (gpr41)*; FFAR3: *free fatty acid receptor 2 (gpr43)*; ADRA2C: *alpha 2 adrenergic receptors subtype C*; PPYR1: *pancreatic polypeptide receptor 1*; CCKBR: *cholecystokinin receptor 4*; IL-25: *interleukin-25*; TLR4: *pattern recognition receptors, such as Toll-like receptor 4*; TNFa: *tumor necrosis factor alpha*; B-Def: *beta-defensin*.

**Table 1 animals-12-01972-t001:** Ingredient and nutrient composition of the dietary concentrates.

Item	Growing ^1^	Finishing ^2^
Ingredient, g/kg		
Corn grain meal	399.7	436.9
Barley grain meal	179.8	150.2
DDGs	179.8	150.2
Wheat	109.7	109.8
Beet pulp	73.9	80.0
Palm oil	20.0	45.0
Calcium carbonate	15.5	12.8
Urea	8.0	4.0
Sodium bicarbonate	5.0	4.0
Dicalcium phosphate	3.6	3.1
Vitamin premix	3.0	2.0
Salt	2.0	2.0
Nutrient		
ME, Mcal/kg DM	3.21	3.29
CP, g/kg DM	157	136
Ether extract, g/kg DM	58	84
Ash, g/kg DM	56	46
NFD, g/kg DM	178	169
NFC, g/kg DM	551	>565

^1^ From 0 to 112 days of the study. ^2^ From 113 days to the end of the study.

**Table 2 animals-12-01972-t002:** Performance and concentrate intake for growing and finishing phase, and for the whole study in Holstein bulls fed high-concentrate diets supplemented with citrus flavonoids.

	Treatment ^1^		*p*-Value ^2^			
Item	Control	BF	SEM	T	Time	T × Time
Growing phase						
Initial age, d	144.9	147.2	0.60	*		
Final age, d	256.9	259.2	0.67	*		
Initial BW, kg	178.2	178.1	6.64	NS		
Final BW, 112 d of study,						
kg	349.2	349.8	9.27	NS		
CV, %	7.9	8.1	0.75	NS		
ADG,						
kg/d	1.53	1.53	0.017	NS	***	NS
CV, %	23.9	25.7	0.83	NS	NS	NS
Concentrate DM intake,						
kg/d	6.09	5.76	0.116	*	***	NS
CV, %	14.5	15.2	0.93	NS	***	NS
FCR, kg/kg	4.68	4.60	0.196	NS	NS	NS
Finishing phase						
Initial age, d	256.9	259.2	0.67	*		
Final age, d	312.9	315.1	0.62	*		
Initial BW, kg	349.2	349.9	9.27	NS		
Final BW, 168 d of study,						
kg	439.6	440.9	9.95	NS		
CV, %	8.0	8.1	0.56	NS		
ADG,						
kg/d	1.62	1.63	0.034	NS	NS	NS
CV, %	32.4	28.7	2.01	NS	NS	NS
Concentrate DM intake,						
kg/d	7.46	7.32	0.194	NS	***	NS
CV, %	13.0	17.0	1.67	NS	**	NS
FCR, kg/kg	4.68	4.60	0.196	NS	NS	NS
Whole						
Initial age, d	144.8	147.2	0.60	*		
Final age, d	312.8	315.6	0.62	*		
Initial BW, kg	178.2	178.13	6.64	NS		
Final BW, 168 d of study,						
kg	439.6	440.9	9.95	NS		
CV, %	8.0	8.1	0.56	NS		
ADG,						
kg/d	1.56	1.56	0.018	NS	***	NS
CV, %	26.7	26.7	0.85	NS	**	NS
Concentrate DM intake,				NS		
kg/d	6.55	6.28	0.147	NS	***	NS
CV, %	14.0	15.8	1.09	NS	***	NS
FCR, kg/kg	4.21	4.05	0.117	NS	***	NS

^1^ C = nonsupplemented, BF = concentrate supplemented with citrus flavonoids at 0.04%. ^2^ T = treatment effect. Time = time effect (period of 14 d). T × Time = treatment by time interaction effect. *** = *p* < 0.001; ** = *p* < 0.01; * = *p* < 0.05.

**Table 3 animals-12-01972-t003:** Carcass quality from Holstein bulls fed high-concentrate diets supplemented with citrus flavonoids.

	Treatment ^1^			*p*-Value ^2^
Item	C	BF	SEM	T
Age before slaughter, d	331.4	333.5	0.74	*
Days in study, d	186.5	186.4	0.43	NS
BW before slaughter, kg	460.7	460.6	3.44	NS
Hot carcass weight, kg	243.8	244.1	2.07	NS
Dressing percentage, %	52.91	53.00	0.255	NS
Fatness, %				NS
2	30.14	41.67		
3	69.86	58.33		
Conformation, %				NS
P	65.75	68.06		
O	32.88	29.17		
R	1.37	2.78		

^1^ C = nonsupplemented, BF = concentrate supplemented with citrus flavonoids at 0.04%. ^2^ T = treatment effect. * = *p* < 0.05.

**Table 4 animals-12-01972-t004:** General activities (%) and social behavior (times/15 min) for growing phase in Holstein bulls fed high-concentrate diets supplemented with citrus flavonoids.

Item	Treatment ^1^			*p*-Values ^2^		
	Control	BF	SEM ^3^	T	Time	T × Time
General Activity, %						
Standing	75.02	74.64	0.031	NS	***	NS
Lying	24.98	25.36	0.099	NS	***	NS
Eating concentrate	8.80	11.99	0.058	***	***	NS
Eating straw	9.96	14.30	0.062	**	*	NS
Drinking	1.81	1.89	0.003	NS	NS	NS
Ruminating	8.62	13.48	0.092	*	*	NS
Social behavior/15 min						
Self-grooming	21.81	24.14	1.075	NS	***	NS
Social	2.83	2.61	0.667	NS	*	NS
Oral non-nutritive	2.55	0.59	0.210	***	*	NS
Fighting	8.02	3.83	0.917	***	***	NS
Butting	3.19	0.86	0.110	***	***	NS
Displacement	1.17	0.66	0.411	NS	***	t
Chasing	0.95	0.06	0.267	***	NS	NS
Chasing up	0.22	0.05	0.111	*	NS	NS
Flehmen	1.89	1.30	0.146	**	***	NS
Attempt to mount	1.97	0.36	0.622	**	NS	NS
Complete mounts	3.16	1.88	0.935	NS	*	NS

^1^ C = nonsupplemented, BF = concentrate supplemented with citrus flavonoids at 0.04%. ^2^ T = treatment effect. Time = time effect (measurements every 14 d). T × Time = treatment by time interaction *** = *p* < 0.001; ** = *p* < 0.01; * = *p* < 0.05; and t = *p* < 0.10. ^3^ SEM = standard error of the means of the log-transformed data (general activity) or root-transformed data (social behavior).

**Table 5 animals-12-01972-t005:** General activities (%) and social behavior (times/15 min) for finishing phase in Holstein bulls fed high-concentrate diets supplemented with citrus flavonoids.

Item	Treatment ^1^			*p*-Values ^2^		
	Control	BF	SEM ^3^	T	Time	T × Time
General Activity, %						
Standing	62.66	60.90	0.075	NS	***	NS
Lying	37.34	39.10	0.106	NS	***	NS
Eating concentrate	5.56	9.81	0.035	***	NS	NS
Eating straw	5.87	9.89	0.152	NS	NS	NS
Drinking	1.45	1.68	0.006	NS	NS	NS
Ruminating	9.72	15.97	0.126	***	***	NS
Social behavior/15 min						
Self-grooming	11.75	15.25	1.127	**	**	NS
Social	4.04	5.63	0.848	NS	***	t
Oral non-nutritive	3.29	0.92	0.282	***	NS	NS
Fighting	10.46	4.53	0.803	***	***	NS
Butting	5.50	0.92	0.664	***	**	NS
Displacement	1.71	0.17	0.315	***	NS	NS
Chasing	0.61	0.00	0.294	NS	NS	NS
Chasing up	0.08	0.00	0.046	t	NS	NS
Flehmen	4.42	2.08	0.464	***	t	NS
Attempt to mount	2.24	0.58	0.756	t	NS	NS
Complete mounts	2.92	1.25	0.130	t	**	NS

^1^ C = nonsupplemented, BF = concentrate supplemented with citrus flavonoids at 0.04%. ^2^ T = treatment effect. Time = time effect (measurements every 14 d). T × Time = treatment by time interaction. *** = *p* < 0.001; ** = *p* < 0.01; and t = *p* < 0.10. ^3^ SEM = standard error of the means of the log-transformed data (general activity) or root-transformed data (social behavior).

**Table 6 animals-12-01972-t006:** Macroscopical observations of the rumen and liver at the slaughterhouse of Holstein bulls fed high-concentrate diets supplemented with citrus flavonoids.

	Treatment ^1^		*p*-Value ^2^
Item	C	BF	
Color of the rumen ^3^			**
1	9.59	22.22	
2	36.99	50.00	
3	53.42	26.39	
4		1.39	
Papillae clumping			NS
Yes	43.90	40.51	
No	56.16	59.49	
Baldness region			t
Yes	16.44	6.94	
No	83.56	93.06	
Liver abscess ^4^			NS
None	79.45	79.17	
A	6.85	8.33	
A-	8.22	4.17	
A+	1.37		
Inflammation	2.74	8.33	

^1^ C = nonsupplemented, BF = concentrate supplemented with citrus flavonoids at 0.04%. ^2^ T = treatment effect. ** = *p* < 0.01; and t = *p* < 0.10. ^3^ Adapted from González et al. [16]: rumen color, 1 = white and 5 = black. ^4^ Adapted from Brown et al. [19].

**Table 7 animals-12-01972-t007:** Rumen VFA concentration at the slaughterhouse of Holstein bulls fed high-concentrate diets supplemented with citrus flavonoids.

	Treatment ^1^			*p*-Value ^2^
	C	BF	SEM	
pH	6.06	6.57	0.122	**
Total VFA, mM	131.3	96.1	7.62	**
Individual VFA, mol/100 mol				
Acetate	53.6	58.9	1.34	**
Propionate	35.9	29.8	1.56	**
Isobutyrate	0.8	0.9	0.09	NS
*n*-butyrate	6.6	7.2	0.24	*
Isovalerate	1.7	1.7	0.24	NS
Valerate	1.5	1.6	0.09	NS
Acetate:propionate, mol/mol	1.6	2.1	0.12	*

^1^ C = non-supplemented, BF = concentrate supplemented with citrus flavonoids at 0.04%. ^2^ T = treatment effect. ** = *p* < 0.01; * = *p* < 0.05

## Data Availability

Data is contained within the article.

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
