# Peer review of "Supplementing Citrus aurantium Flavonoid Extract in High-Fat Finishing Diets Improves Animal Behavior and Rumen Health and Modifies Rumen and Duodenum Epithelium Gene Expression in Holstein Bulls"

_animals, 2022, doi:10.3390/ani12151972_

Round 1

Reviewer 1 Report

The title of the manuscript needs rewrite. The number of the animals are differed in the abstract than at the materials and methods section. In addition, the authors used growing Holstien calves not bulls.

Author Response

The title of the manuscript needs rewrite.

AA: Sorry, we do not follow the comment, in which sense should we change it? However here our suggestion, as we are not sure why the title was not fulfilling the expectations, we suggest it and after the reviewer’s approval we will change it

Citrus aurantium flavonoid extract supplementation improves animal behavior, rumen health and modifies rumen and duodenum gene expression in bulls fed high-fat finishing diet

The number of the animals are differed in the abstract than at the materials and methods section.

AA: Corrected, thanks

In addition, the authors used growing Holstein calves not bulls.

AA: We consider them bulls as they are sexually active, sexual maturity is achieved around 7 mo of age (, Sexual Maturation in the Bull, N Rawlings,ACO Evans,RK Chandolia,ET Bagu, First published: 09 July 2008. https://doi.org/10.1111/j.1439-0531.2008.01177.x) even during the growing phase the complete mounts is as much as frequent as in the finishing phase (see table 4 and 5). This is the last study of a PhD thesis (Paniagua, M., Crespo, J., Bach, A., Devant, M. Effects of flavonoids extracted from Citrus aurantium on performance, eating and animal behavior, rumen health, and carcass quality in Holstein bulls fed high-concentrate diets. Feed Sci. Technol. 2018, 246, 114–126. DOI:10.1016/j.anifeedsci.2018.08.010.; Paniagua M.; Crespo J.; Arís A.; Devant M. Citrus aurantium flavonoid extract improves concentrate efficiency, animal behavior, and reduces rumen inflammation of Holstein bulls fed high-concentrate diets. Ani. Feed Sci Technol. 2019, 258, 114304. DOI:10.1016/j.anifeedsci.2019.114304.; Paniagua M.; Crespo, F.J.; Arís, A.; Devant, M. Effects of Flavonoids Extracted from Citrus aurantium on Performance, Behavior, and Rumen Gene Expression in Holstein Bulls Fed with High-Concentrate Diets in Pellet Form. Animals 2021, 11, 1387. DOI:10.3390/ani11051387.). In previous published papers the importance of reducing the aggressive and sexual behaviours of bulls have been lighted; therefore, we would like to keep the word “bull” as it better describes the main welfare issues (and meat quality problems) related to their gender.

Reviewer 2 Report

The research question is interesting, considering that the strategy of using feed additives or supplements for the improvement of animal health and welfare, with an integrative approach of the One Health/One Welfare concept, is becoming more and more important in animal production.

The paper is well presented, and the conclusion is supported by the results, although the yield results are only slightly conclusive. In the reviewer's opinion, it is more difficult to work under field conditions with commercial farms.

Nonetheless, in the reviewer's opinion there are several comments and clarifications to be considered by the authors to improve the manuscript’s quality.

Sample summary

Line 28: It is advisable to specify the source of fat (saturated), as the type of fat (saturated/unsaturated) has different effects on the immunomodulatory function of inflammation.

Abstract:

Line 41: What do the numbers (7,16,39) mean? Member?

Qué significan los números (7,16,39)?. Member?

Line 42: gen expression higher?

Introduction

Lines 64-65: The differences found between the two studies may be due to the technological process of pelleting, which may cause some modification in the flavonoid characteristics due to the temperature of the process.

Materials and Methods

Line 109-123: All animals received the same vaccination schedule? Please, include information.

Line 113: one space remains (0.60 d)

Line 115: There was an adaptation phase to the BF treatment before recording of feed consumption data?.

Line 136: 113 d?

Line 137: in abstract fat % concentrate was 84 and here 80 g/Kg DM.  Specify the percentage of fat incorporated in the concentrate.

Line 133: Concentrate consumption?.

Line 140-141: Information repeated

Line 183: 2.3? and so on.

Line 177: 10:30 am

Line 177-182: Please, extend the information on the frequency of recording of general activities and social behaviour. The frequency of recording data of body weight is also not reported.

Line 184: HCW (hot carcass weight)

Line 184-192: When were the animals slaughtered?

Line 200-202: How many animals were sampled for each treatment and pen?  It is not clear.

Line 228: rumen or ruminal papillae?

Line  232-233: The RNA integrity should be asses with an bioanalyzer equipment, for example.  Please include the criteria used to consider the quality of the extracted RNA as satisfactory.

Line 247: Where is table 3?. Table 3 is related to Carcass quality (line 312).

Line 259-260: Eq. [3.5].It is ok?.

Results

Line 295-298:  Only one reference to Table 2. 

Table 2: Initial BW, kg    178.2    178.13 6.64      NS . // 178.13 should be 178.1

Line 298-302: If the results are not statistically significant and are also an estimation, they should not be included in the manuscript.

Table 3: According to the age at slaughter data, the animals were slaughtered 18 days after the end of the finishing period. Consumption was maintained for both treatments from day 312.8 to 331.4 for treatment C and from day 315.6 to 333.5 for treatment BF? It is important that this information should be clarified, so that the rumen and duodenum samples were taken while the two diets, C and BF, were being consumed.

In table 3, in the fatness % variable, rank 1 is indicated (no data) but apparently, rank 4 and 5 also have no data. Please, improve the presentation of the results in the table.

In the variable conformation %, it is noteworthy that the highest percentage of the carcasses have a conformation "p" poor conformation and, on the other hand, a fat cover level of 3.

In tables 4 and 5, the sum of the overall activities adds up to more than 100%. Is this correct?.

Line 331, 373, 411: high-concentrate

Table 6: Superscripts 3 and 4 not described.

In figure 1, the difference in TAS2R28 gene expression of the two treatments appears to be statistically significant, although the text indicates that it is not.

In the results related to the expression study, how can the authors explain the difference in scale (and therefore expression) of the genes studied depending on whether the sample is rumen papilla or duodenum? And the under-expression (negative values) of certain genes in the duodenum sample.

Discussion

In general, the discussion does not refer to the type of fat (saturated and unsaturated) which affects the ruminal biohydrogenation process and ruminal microbiota in a different way. In the formulation of diets to incorporate fat, both the level of inclusion and the type of fat must be considered.

At some points, the discussion is not very precise and reference is made to analyses that have not been carried out, for example on the rumen microbiota and the relative abundance of certain bacteria such as species of Butyrivibrio spp.

Line 543: (Mielenz, 2016)[36]: Delete the text of the reference and checks traceability. It appears to be reference 40?.

Line 555-560:

Line: 565: additional

Line 699: missing punctuation mark

Author Response

The research question is interesting, considering that the strategy of using feed additives or supplements for the improvement of animal health and welfare, with an integrative approach of the One Health/One Welfare concept, is becoming more and more important in animal production.

The paper is well presented, and the conclusion is supported by the results, although the yield results are only slightly conclusive. In the reviewer's opinion, it is more difficult to work under field conditions with commercial farms.

Nonetheless, in the reviewer's opinion there are several comments and clarifications to be considered by the authors to improve the manuscript’s quality.

Sample summary

Line 28: It is advisable to specify the source of fat (saturated), as the type of fat (saturated/unsaturated) has different effects on the immunomodulatory function of inflammation.

AA: The reviewer is right, type of fat may affect immunomodulatory function or inflammation. Usually PUFA (and the type of PUFA) have a greater impact in gut inflammatory status than saturated fats (efaidnbmnnnibpcajpcglclefindmkaj/https://jasbsci.biomedcentral.com/track/pdf/10.1186/s40104-022-00690-7.pdf), in the present study fat was increased by increasing palm oil inclusion, a fat source with a low percentge of PUFA. In addition, in ruminants rumen FA biohydronation decreases the amount of PUFA interacting with the gut epithelia. Therefore, it has not been discussed their potential role in gut inflammation.

But more importantly, in the present study, as indicated in the introduction and discussed later, the aim of the present study was to evaluate the supplementation of BF in diets with high fat inclusion levels and the potential additive effects of BF supplementation with high-fat diets on feed intake. Most common fat source used for finishing cattle is palm oil. As indicated increasing dietary fat content at the finishing phase has been a strategy to fulfill animals energy to modulate feed intake and reduce the meal size and to control bloat and rumen acidosis problems. The mode of action of fat on intake regulation (serum CCK and PP increases) may interfere with the mode of action observed with BF supplementation. In the simple summary (line 26) there is no room for this discussion. Potential interactions between fat supplementation and BF mode of action have been widely discussed from line 527 to 548.

Abstract:

Line 41: What do the numbers (7,16,39) mean?

AA: They correspond to the bitter receptor subtypes.  Bitter taste is recognized by a subfamily of G-protein coupled receptors (GPCRs), called the T2Rs or TAS2Rs. The number of subtypes varies in different species (from 3 in chicken to 50 in frog).

Line 42: gen expression higher?

AA: Substituted greater by higher?

Introduction

Lines 64-65: The differences found between the two studies may be due to the technological process of pelleting, which may cause some modification in the flavonoid characteristics due to the temperature of the process.

AA: Right, it has been discussed previously in Paniagua M.; Crespo, F.J.; Arís, A.; Devant, M. Effects of Flavonoids Extracted from Citrus aurantium on Performance, Behavior, and Rumen Gene Expression in Holstein Bulls Fed with High-Concentrate Diets in Pellet Form. Animals 2021, 11, 1387. DOI:10.3390/ani11051387.

Materials and Methods

Line 109-123: All animals received the same vaccination schedule? Please, include information.

AA: During the study animals were not vaccinated, it has been indicated

Line 113: one space remains (0.60 d)

AA: corrected

Line 115: There was an adaptation phase to the BF treatment before recording of feed consumption data?.

AA: No, there was no adaptation phase to BF

Line 136: 113 d?

AA: 8 periods of 14 days is 112 d, we consider the first day of the study as day 0, so yes, during the first 112 days

Line 137: in abstract fat % concentrate was 84 and here 80 g/Kg DM.  Specify the percentage of fat incorporated in the concentrate.

AA: Corrected, it said around 80 g/d, we have indicated 84 to be more precised

Line 133: Concentrate consumption?.Line 140-141: Information repeated

AA: corrected

Line 183: 2.3? and so on.

AA: The subtitle of Animal behavior was missing, it has been corrected

Line 177: 10:30 am

AA: corrected

Line 177-182: Please, extend the information on the frequency of recording of general activities and social behaviour. The frequency of recording data of body weight is also not reported.

AA: corrected

Line 184: HCW (hot carcass weight)

AA: corrected

Line 184-192: When were the animals slaughtered?

AA: Indicated in the new version

Line 200-202: How many animals were sampled for each treatment and pen?  It is not clear.

AA: In the new version it has been indicated:

In all animals rumen and liver of every animal were macroscopically evaluated at the slaughterhouse.

And the exact number has more clearly indicated :

Additionally, a liquid sample from rumen was obtained from homogeneous contents strained with a cheesecloth from 18 animals per treatment randomly selected from two pens, immediately following slaughter.

Line 228: rumen or ruminal papillae?

Line  232-233: The RNA integrity should be asses with an bioanalyzer equipment, for example.  Please include the criteria used to consider the quality of the extracted RNA as satisfactory.

AA: It has been indicated. The criteria used to consider a good RNA quality was a ratio 260/280 and 260/230 in the range of ~2.0   and 2.0–2.2, respectively.

Line 247: Where is table 3?. Table 3 is related to Carcass quality (line 312).

AA: Right, it’s a mistake, we have referenced the paper were the data are indicated.

Line 259-260: Eq. [3.5].It is ok?.

AA: Yes, in this paper you can find the equation

Results

Line 295-298:  Only one reference to Table 2. 

AA: corrected

Table 2: Initial BW, kg    178.2    178.13 6.64      NS . // 178.13 should be 178.1

AA: corrected

Line 298-302: If the results are not statistically significant and are also an estimation, they should not be included in the manuscript.

AA: We would like to keep these data; first of all, they give an approximate value of the straw intake and the importance of concentrate intake in the overall feed consumption. They also indicate that as animal grow their percentage of straw intake compared to concentrate intake increases. We are very cautious with these data as we know that animals potentially also can consume straw from the bedding material (mainly the first days when they are provided new straw for bedding), but these data are very valuable from a practical point of view and also indicate the importance of straw consumption in the overall feed consumption.

Table 3: According to the age at slaughter data, the animals were slaughtered 18 days after the end of the finishing period. Consumption was maintained for both treatments from day 312.8 to 331.4 for treatment C and from day 315.6 to 333.5 for treatment BF? It is important that this information should be clarified, so that the rumen and duodenum samples were taken while the two diets, C and BF, were being consumed.

AA: Yes, treatments were maintained throughtout the study until slaughter; it has been described “The study was divided into growing (0 to 112 d) and finishing (113 to 168 d) phase, after 168 animals were fed the same treatments until slaughter day”.

In table 3, in the fatness % variable, rank 1 is indicated (no data) but apparently, rank 4 and 5 also have no data. Please, improve the presentation of the results in the table.

AA: Rank 1 has been deleted

In the variable conformation %, it is noteworthy that the highest percentage of the carcasses have a conformation "p" poor conformation and, on the other hand, a fat cover level of 3.

AA: Yes, we do not follow the reviewer’s comment. Carcass classification observed is common in this type of animals, and it’s highly dependent on the slaughterhouse.

In tables 4 and 5, the sum of the overall activities adds up to more than 100%. Is this correct?.

AA: Yes, only standing and lying behaviors should sum 100%, an animal can be lying and ruminating at the same time.

Line 331, 373, 411: high-concentrate

AA: corrected

Table 6: Superscripts 3 and 4 not described.

AA: Added

In figure 1, the difference in TAS2R38 gene expression of the two treatments appears to be statistically significant, although the text indicates that

AA: We had a mistake in the error bars, it has been corrected-

In the results related to the expression study, how can the authors explain the difference in scale (and therefore expression) of the genes studied depending on whether the sample is

AA: Yes, the gut site affects the scale of gene expression, probably it’s related of the activity/metabolism of the gut site (see for example: https://doi.org/10.2527/jas.53793)

And the under-expression (negative values) of certain genes in the duodenum

AA: Thanks for the comment. There was a mistake in the graph and we have corrected it.

Discussion

In general, the discussion does not refer to the type of fat (saturated and unsaturated) which affects the ruminal biohydrogenation process and ruminal microbiota in a different way. In the formulation of diets to incorporate fat, both the level of inclusion and the type of fat must be considered.

AA:  See previous comments. And as mentioned previously, type of fat has not been discussed because most common fat source in finishing diets is palm oil (because of it’s price)

At some points, the discussion is not very precise and reference is made to analyses that have not been carried out, for example on the rumen microbiota and the relative abundance of certain bacteria such as species of Butyrivibrio spp.

AA:  We may agree that there is some level of speculation, but we think that in the discussion authors may have the freedom to debate of potential explanations if they are cautious.

Line 543: (Mielenz, 2016)[36]: Delete the text of the reference and checks traceability. It appears to be reference 40?.

AA:  Corrected

Line 555-560:?

AA:  Sorry, we do not follow reviewer’s comment

Line: 565: additional

AA:  Corrected

Line 699: missing punctuation mark

AA:  Corrected

Reviewer 3 Report

Dear Author,

the present research is interesting and well written. However, the manuscript has many similarities both in the materials and methods and in the results sections with previous manuscript published by the same authors. Therefore, it cannot be accepted for publication in the present form and a detailed review in materials and methods and results sections should be done.

Minor comments are indicated in attached pdf file.

Author Response

Dear Author,

the present research is interesting and well written. However, the manuscript has many similarities both in the materials and methods and in the results sections with previous manuscript published by the same authors. Therefore, it cannot be accepted for publication in the present form and a detailed review in materials and methods and results sections should be done.

AA:  This manuscript is part of Dr. Paniagua thesis and some experiments share materials and methods sections, therefore it has similarities. Sometimes, trying to avoid similarities we reword or rephrase paragraph, but it’s quite difficult to do it without incurring in wrong descriptions. It should be accepted that materials and methods sections have similarities, we should be able to reproduce the studies.

PDF

Return the values ​​with two digits after the comma. Please, check the whole manuscript.

AA:  Sorry we do not follow the comment

Straw description-This part might be insert in 2.1. section.

AA:  Rephrased

Add in Table 1 bitter orange extract concentration.

AA:  The bitter orange concentration corresponds to 0.04 % of bitter orange extract (Citrus aurantium) of the whole fruit rich in naringin (24%) (Bioflavex CA, HTBA, S.L.U., Barcelona, Spain). We haven’t added it in order to avoid a table like the one below that duplicates information.

Item.

Growing1

Finishing2

Ingredient, g/ kg

Control3

BF3

Control3

BF3

Corn grain meal

399.7

399.3

436.9

436.5

Barley grain meal

179.8

179.8

150.2

150.2

DDGs

179.8

179.8

150.2

150.2

Wheat

109.7

109.7

109.8

109.8

Beet pulp

73.9

73.9

80.0

80.0

Palm oil

20.0

20.0

45.0

45.0

Calcium carbonate

15.5

15.5

12.8

12.8

Urea

8.0

8.0

4.0

4.0

Sodium bicarbonate

5.0

5.0

4.0

4.0

Dicalcium phosphate

3.6

3.6

3.1

3.1

Vitamin premix

3.0

3.0

2.0

2.0

Salt

2.0

2.0

2.0

2.0

Bioflavex

0.4

0.4

Nutrient

ME, Mcal/kg DM

3.21

3.21

3.29

3.29

CP, g/ kg DM

157

157

136

136

Ether extract, g/ kg DM

58

58

84

84

Ash, g/ kg DM

56

56

46

46

NFD, g/ kg DM

178

178

169

169

NFC, g/ kg DM

551

551

565

565

Naringin, g/kg

0.096

0.096

1 from 0 to 112 days of the study.

2 from 113 days to the end of the study.

3 C = non-supplemented, BF = concentrate supplemented with citrus flavonoids at 0.04%.

Table 2-& Table 3 t = P < 0.10 is not reported in Table. Delete.

AA:  corrected

Table 7- Add **= P < 0.01; * = 0.05 in Table caption.

AA:  added

Footnote figure 2- not clear

AA:  We do not follow the reviewer’s comment, what’s not clear?

Round 2

Reviewer 2 Report

  • Thank you for replying to the reviewer's comments. I agree with you on the majority, but I miss more information on the quality of the RNA (RIN values >6, for example if it has been evaluated) and more argumentation in the discussion of results, especially those concerning the microbiota data. 

Reviewer 3 Report

The authors have satisfactorily responded to all my questions and made the necessary changes to the manuscript.